# The Associations among Gender, Age, eHealth Literacy, Beliefs about Medicines and Medication Adherence among Elementary and Secondary School Teachers

**DOI:** 10.3390/ijerph19116926

**Published:** 2022-06-06

**Authors:** Chiao Ling Huang, Chia Hsing Chiang, Shu Ching Yang, Fu-Zong Wu

**Affiliations:** 1Department of Educational Information Technology, Faculty of Education, East China Normal University, Shanghai 200062, China; leekbox@gmail.com; 2Intelligent Electronic Commerce Research Center, Institute of Education, National Sun Yat-Sen University, Kaohsiung 80424, Taiwan; d996050002@gmail.com; 3Department of Radiology, Kaohsiung Veterans General Hospital, Kaohsiung 813414, Taiwan; 4Department of Post-Baccalaureate Medicine, National Sun Yat-Sen University, Kaohsiung 813414, Taiwan

**Keywords:** beliefs about medicines, eHealth literacy, medication adherence, teacher

## Abstract

*Background:* A lack of health literacy may negatively impact patient adherence behavior in health care delivery, leading to a major threat to individual health and wellbeing and an increasing financial burden on national healthcare systems. Therefore, how to cultivate citizens’ health literacy, especially electronic health (eHealth) literacy that is closely related to the Internet, may be seen as a way to reduce the financial burden of the national healthcare systems, which is the responsibility of every citizen. However, previous studies on medication adherence have mostly been conducted with chronic disease patient samples rather than normal samples. Teachers are not only the main body of school health efforts, but also role models for students’ healthy behavior. Therefore, understanding differences in eHealth literacy beliefs among schoolteachers would be helpful for improving the existing health promoting programs and merit specific research. *Aims:* The present study identified the relationships among gender, age, electronic health (eHealth) literacy, beliefs about medicines, and medication adherence among elementary and secondary school teachers. *Methods:* A total of 485 teachers aged 22–51 years completed a pen-and-paper questionnaire. The instruments included an eHealth literacy scale, a belief about medicines scale and a medication adherence scale. *Results:* The results showed a significant difference between genders in necessity beliefs about medication (*t* = 2.00, *p* < 0.05), and a significant difference between ages in functional eHealth literacy (*F* = 3.18, *p* < 0.05) and in necessity beliefs about medication (Welch = 7.63, *p* < 0.01). Moreover, age (*β* = 0.09), functional eHealth literacy (*β* = 0.12), and necessity beliefs about medication (*β* = 0.11) positively predicted medication adherence, while concerns about medication (*β* = −0.23) negatively predicted medication adherence. *Conclusions:* The results showed that male teachers had stronger concerns about medication than female teachers. Teachers aged 42–51 years had lower functional eHealth literacy and stronger necessity beliefs about medication than teachers aged 22–31 years. In addition, teachers who were older, had higher functional eHealth literacy, had stronger necessity beliefs about medication, and had fewer concerns about medication tended to take their medications as prescribed. These findings revealed that helping teachers develop high eHealth literacy and positive beliefs about medicines is an effective strategy for improving medication adherence.

## 1. Introduction

Medication adherence is usually defined as whether patients follow the prescription to take medicine, accept therapy, and continuously take a prescribed medication until they are cured or stop when the condition is sufficiently improved [1]. Patient adherence in healthcare services affects personal health and well-being, as well as the financial affordability of the national healthcare system [2]. Patient adherence can be seen as a way to reduce the financial burden on the national health care system and help the country’s medical economy. However, patients’ medication adherence is intertwined with many factors, including their medication beliefs, which are defined as an individual’s beliefs about the necessity of and problems with medication [3,4].

The meta-analytic review by Horne et al. regarding the necessity-concerns framework for understanding patients’ adherence-related beliefs about medicines prescribed for long-term conditions pointed out that the ‘Necessity-Concerns Framework (NCF)’ potentially offers a convenient model for understanding patients’ perspectives on prescribed medicines, including perceptions of personal need for treatment (necessity beliefs) and concerns about a range of potential adverse consequences [5]. Horne et al. (2013) mentioned that nonadherence is often a hidden problem that patients may not be willing to express doubts or concerns about prescribed medication and to report nonadherence sometimes because they fear that this will be perceived by the prescriber as a lack of faith in them [5]. Prior studies have shown that greater adherence is linked to fewer concerns about medication and stronger necessity beliefs about medication [6,7,8,9]. Thus, patients with positive perspectives on medicines are more likely than those with negative perspectives to take their medications as prescribed.

Health literacy is an individual’s set of abilities to obtain, realize, and utilize health information, including disease and medication knowledge [10]. Recently, the internet has been widely used to access and share health information, resulting in increased attention to electronic health (eHealth) literacy. Norman et al. [11] defined eHealth literacy as “the ability to seek, find, understand, and appraise health information from electronic sources and apply the knowledge gained to address or solve a health problem”. Moreover, eHealth offers an opportunity to help patients better manage their medications and thus leads to better medication adherence [12]. 

Studies have demonstrated that patients with higher health literacy tend to have greater adherence [13,14,15]. For example, Persell et al. [16] found that patient health literacy was strongly associated with knowledge of medicine indications and understanding of instructions and dosing, implying that low health literacy was associated with worse medication self-management in many aspects. Lu and Zhang [17] investigated the association between eHealth literacy in online health communities (OHCs) and patient adherence based on social cognitive theory. Their results showed that patients’ eHealth literacy was positively correlated with patient adherence through the mediations of physician–patient communication, internet health information-seeking behavior, and perceived quality of internet health information in OHCs, which implied that physicians can improve their patients’ eHealth literacy to facilitate treatment more efficiently. In this regard, given that most of the existing research focuses on the relationship between the functional definition of health literacy and medication [13,14,15,16], this study provides a more comprehensive perspective on eHealth literacy by including the proactive role of interactive and critical eHealth literacy in one’s own health by leveraging internet technology.

Notably, for chronic disease patients, individual factors such as gender and age have been shown to act as predictors of medication adherence. In this regard, male patients have greater adherence than female patients [13,18]. However, for kidney transplant recipients aged 11–16 years, no gender difference in adherence has been observed, while females have higher adherence than males among participants 17–24 years old [19]. Other studies have also discovered that older adults have greater adherence than younger adults [18,20]. For the dissimilar findings of extant literature, the systematic review by Gast et al. [21] also showed that the direction of the influence of gender on medication adherence was not consistent, and that age seemed to have a convex relationship with medication adherence, indicating lower adherence in younger and older people. Therefore, considering eHealth literacy, medication beliefs, and personal characteristics simultaneously will obtain more comprehensive and complete outcomes when discussing an individual’s medication adherence.

In Taiwan, individuals can quickly and conveniently access information about medication and diseases from the internet. Since 1995, the National Health Insurance system has enabled people to access a comprehensive set of medical services and obtain many medications from different medical institutions in Taiwan [22]. If patients follow their prescribed treatments reasonably closely, they benefit from the effectiveness of their medications [23]. In contrast, poor medication adherence leads to increased risks of morbidity and mortality [24]. Therefore, the Health Promoting School (HPS) program is devoted to promoting safe medication use that enhances students’ and teachers’ correct medication-taking behaviors [22]; in other words, the HPS program has focused its interest and attention on school personnel. As the HPS program suggests, schools should take responsibility to care for students and make them become the valuable human resources assets of society. In this regard, schools need to actively increase their health efforts to properly utilize various health-promoting opportunities. To accomplish this goal, we argue that equal attention should be given to teachers (providers) because teachers’ health skills and medication adherence may be modeled by or transferred to their school students.

The majority of Taiwanese elementary and secondary school teachers can search for and understand disease and medication knowledge from internet resources, leading to good decisions about their medicine management. However, little is known about the associations among teachers’ gender, age, eHealth literacy, specific beliefs about medicines and medication adherence. 

Taiwanese school-aged children spend 8–9 hours a day in schools, and teachers became critical life instructors and mentors during that time. In terms of teaching health education, teachers are not only the main body of school health efforts, but also role models for students’ healthy behavior. Thus, we suggest that improving teachers’ health-promoting behaviors will induce students’ healthy behaviors and produce positive health outcomes. As prior studies have supported, educator health literacy and the ability to support student health are associated with school teacher-level characteristics [25,26]; hence, it is important to understand differences in eHealth literacy beliefs among primary and secondary school teachers.

Moreover, previous studies on medication adherence have mostly been conducted based on chronic disease patient samples rather than normal samples, resulting in a research gap. To obtain a better understanding of teachers’ medication adherence, teachers, used as a research sample, are a worthy subject of study. Based on the present practical applications and research gaps, this paper aims to investigate the associations between teachers’ individual factors, eHealth literacy, and medication adherence, regarding which meaningful research questions can be posed. Three hypotheses are proposed based on the extant literature:

**Hypothesis** **1** **(H1).**
*Teachers of different genders have different eHealth literacy levels, medication beliefs and medication adherence levels.*


**Hypothesis** **2** **(H2).**
*Teachers in different age groups have different eHealth literacy levels, medication beliefs and medication adherence levels.*


**Hypothesis** **3** **(H3).**
*Individual factors (i.e., gender and age), eHealth literacy and medication beliefs can jointly predict medication adherence.*


## 2. Materials and Methods

### 2.1. Study Design and Participants

This cross-sectional study was conducted at elementary and secondary schools in Taiwan. To participate in this study, the subjects had to have teacher status and volunteer to participate. Those on extended leave (e.g., parental leave) were not included in this study.

#### 2.1.1. Pretesting

A total of three scales were used in this study, two of which were designed by the authors, and pretest samples were mainly used to examine these two scales. The appropriateness of the scale questions and response options was evaluated by three specialist professors and three elementary and secondary school teachers at the beginning stage of scale development. The researchers ensured that the questions in the survey were clearly articulated based on the comments provided by the invited teachers and professors, and that the response options were relevant and mutually exclusive, in addition to assessing the response latency.

Pretesting was performed to address item reduction, factor extraction and reliability analysis. According to the guidance of researchers [27,28], the appropriate sample size for a scale factor analysis is more than 120. Therefore, a total of 150 paper-and-pen questionnaires (self-reported type) were distributed, of which 127 were valid. All questions in both scales remained unchanged because the results of the analyses met the requirements.

#### 2.1.2. Full-Scale Administration

According to the Ministry of Education in Taiwan [29], in 2019, for elementary schools, the population of male teachers was 27,414 (28.7%), and that of female teachers was 68,256 (71.3%). For secondary schools, the population of male teachers was 14,366 (30.9%), and that of female teachers was 32,086 (69.1%). As a whole, the population of male teachers in both elementary and secondary schools was 41,780 (29.4%), and that of female teachers was 100,342 (70.6%). That is, the ratio of male to female teachers in elementary and secondary schools was approximately 3 to 7; thus, the sampling ratio of male and female participants in the full-scale administration stage was also set to 3 to 7, enabling us to be in line with the current gender ratio of the teacher population.

In addition, given that Taiwan can be roughly divided into three regions (northern, central, and southern); we recruited samples from these three regions and determined that at least 100 teachers from each region were needed. Finally, a convenience sample of six hundred questionnaires was distributed, and 485 questionnaires were returned, resulting in a response rate of 80.8%.

Among the 485 participants, 209 (43.1%) were elementary school teachers, 273 (56.3%) were secondary school teachers, and three (0.6%) did not specify their status. A total of 178 participants (36.7%) taught in the southern region of Taiwan, 114 (23.5%) taught in the northern region, and 193 (39.8%) taught in the central region. The participants’ mean (*SD*) age was 39.55 (6.58) years. Of the 485 participants, 348 were female, 137 were male, and the proportion of the sample collected basically met our expectations.

Notably, teachers in Taiwan can retire after 25 years of service. Thus, we divided the collected data into three groups based on the age of the teachers (using 10 years for interval grouping), which helped to compare teachers with different levels of seniority. Teachers aged 22–31 were categorized into the first group (G1), those aged 32–41 were categorized into the second group (G2), and those aged 42–51 were categorized into the third group (G3). However, since this was a voluntary study, it was relatively difficult to recruit the same number of subjects per group based on age, and we were unable to do so. In addition, this is a pilot study that aims to understand the current status of the teacher population. Therefore, we used a more general method to evaluate our participants, including those with nonchronic diseases, and did not limit the investigation to one disease/treatment.

### 2.2. Instruments

The instrument employed in this study has four parts: basic personal information, an eHealth literacy scale, a belief about medicines scale, and a medication adherence scale. A total of 127 data points were used to evaluate the validity and reliability of the beliefs about medicines scale and medication adherence scale, which we developed, and the reliability of the eHealth literacy scale. Specifically, we followed the guidance of Boateng et al. [30] to test these scales. In the item development phase, we used the process of concept clarification and invited three specialist professors and three elementary and secondary school teachers to test the content validity of the items that we had generated. These experts also helped us ensure that the questions and response options were meaningful and avoided biased answers from the participants.

In the scale development phase, we determined the pretest sample size based on Comrey’s suggestion [28] and distributed the questionnaire to our pretest sample. The collected data were used to conduct item reduction and factor extraction by performing item analyses and exploratory factor analysis (EFA), and the dimensions were also determined at this stage. 

In the scale evaluation phase, we calculated the coefficient of internal consistency with our pretest data, and the results revealed that all three scales were reliable. Notably, given that the aim of the present study was not the development and validation of measurements, we did not test the criterion-related validity of these two scales, and the detailed validation of these scales warrants study.

#### 2.2.1. eHealth Literacy

Electronic health literacy was assessed based on the eHealth literacy scale (eHLS) developed by Chiang et al. [31]. The eHLS measures participants’ functional, interactive, and critical eHealth literacy. According to Chiang et al.’s report, the validity and reliability of the eHLS were validated using item analyses, EFA and confirmatory factor analysis (CFA), which indicated that the eHLS is a reliable and validated measure. The 12-item eHLS includes the following three aspects: functional (three items), interactive (four items) and critical (five items) literacy. 

The eHLS was rated by the respondents on a 5-point Likert-type scale ranging from 1 “strong disagreement” to 5 “strong agreement”. The total score ranges were between 3 and 15 for functional literacy, between 4 and 20 for interactive literacy, between 5 and 25 for critical literacy, and between 12 and 60 for the total eHLS, with higher scores representing higher eHealth literacy. The internal consistency of the eHLS based on Cronbach’s alpha was 0.84 (functional), 0.90 (interactive) and 0.92 (critical) within the pretest sample.

#### 2.2.2. Beliefs about Medicines

An eight-item beliefs about medicines scale (BMS) was used in this study to measure individuals’ perspectives on the necessity of and problems with medication (focusing on the prescription medicine). We designed this scale on the basis of Horne et al.’s [10] scale. The BMS includes the following two aspects: specific necessity beliefs (four items) and specific concern beliefs (four items). It was rated by the respondents on a 5-point Likert-type scale ranging from 1 “strong disagreement” to 5 “strong agreement”. The total score range was 8–40 for the whole BMS and 4–20 for each of the two aspects, with higher scores representing stronger perceptions of the necessity of medication and stronger concerns about medication.

An EFA (principal axis factor method with direct oblimin rotation) of the pretest sample indicated that the Kaiser–Meyer–Olkin value of the BMS was 0.73, Bartlett’s test of sphericity was significant (*p* < 0.001), the explained variance was 54.0%, and the factor loadings ranged between 0.56 and 0.85. The results of the critical ratio (all t-values > 3) and the correlations between individual questions and total scale scores (all r coefficients > 0.40) met the requirements. The Cronbach’s alpha obtained from the pretest sample was 0.79 for specific necessity beliefs and 0.83 for specific concern beliefs.

#### 2.2.3. Medication Adherence

We administered a medication adherence scale (MAS) containing one aspect to measure the extent to which an individual took medication as prescribed by his or her health care providers (e.g., ear, nose, and throat doctor; dermatologist; or gastrologist) in the past year. Although many patient-based medication adherence questionnaires use one-month as a time interval, in the general population, the number of illnesses and the number of medications taken vary from person to person. In particular, subjects may not have been sick or taking medication in the last week or month. Therefore, it seems difficult to reflect the real situation if the short time intervals are used. Therefore, we consulted the literature on healthcare utilization [32,33] and decided to evaluate teachers’ medication in the past year.

We designed the eight-item MAS based on a thorough review of the literature [34,35]. An EFA with direct oblimin rotation of the pretest sample indicated that the Kaiser–Meyer–Olkin value was 0.89, Bartlett’s test of sphericity was significant (*p* < 0.001), the explained variance was 58.6%, and the factor loadings ranged between 0.61 and 0.82. The results of the critical ratio (all t-values > 3) and the correlations between individual questions and total scale scores (all r coefficients > 0.40) met the requirements. The Cronbach’s alpha value obtained from the pretest sample of the MAS was 0.92. The MAS was rated by the respondents on a five-point Likert-type scale ranging from 1 “never” to 5 “always”. The total score range was between 8 and 40, with higher scores representing greater adherence.

### 2.3. Data Collection

In the pretest (March 2019 to April 2019), we used a convenience sampling technique to recruit subjects for this study. We contacted acquaintances who teach at elementary and secondary schools in Kaohsiung (southern region) to invite them and their coworkers to participate in this study. In the full-scale administration stage (May 2019 to June 2019), we also used a convenience sampling technique to collect data. We contacted teachers who taught at the selected schools and asked whether they would like to assist us in promoting this research and distributing survey questionnaires (self-report paper-and-pen questionnaires). After distributing 600 questionnaires to 23 schools, the response rate was 80.8%.

### 2.4. Ethical Issues

In view of personal privacy and anonymous surveys, the first page of our questionnaire states that filling in the questionnaire and submitting it is deemed consent. The first page of the questionnaire also contained the contact information of the research team, and the participants could contact us if they had questions or needed help in case of confusion when completing the questionnaire. The survey instructions clearly informed the teacher-participants of the purposes of the research and of their rights regarding joining or dropping out of this study. Teachers who are interested in this study have plenty of time to think and decide whether to participate in the investigation after reading the research purposes and related information on the front page of the survey. Those who are willing to engage in this study can return the questionnaire on the agreed time. At the same time, two checkpoints were provided in the research design so that the respondents could stop at any time: (1) As they were completing the survey, the respondents could quit at any time. (2) Upon completing the entire survey, the respondents determined whether to hand it in. The respondents were assured that their participation was voluntary, anonymous, and strictly confidential, and that they had the right to refuse to participate in the study at any time without any penalty. In addition, the teachers who participated in the study received a gift, regardless of whether they completed the questionnaire. Completing the questionnaire took them approximately 10 min. 

### 2.5. Statistical Analysis

To analyze the research data (obtained from the full-scale administration stage), independent t tests and one-way analysis of variance (ANOVA) were used to address the first and second hypotheses, and a linear multiple regression analysis was performed to address the third hypothesis (medication adherence as the dependent variable and gender, age, eHealth literacy and beliefs about medicine as the independent variables). In addition, to understand the central tendency and dispersion of the collected data, descriptive statistical analysis was used to compute the mean, standard deviation, maximum and minimum values. The distribution of the data, including skewness and kurtosis, was examined and found to be normally distributed with acceptable skewness and kurtosis. The statistical significance level was set at 0.05. Notably, since all scales were rated using a 5-point Likert-type format, the cutoff point of the mean value was set to 2.5. A score higher than 2.5 indicated that the individual had better eHealth literacy, medication adherence behavior or stronger beliefs about medicines. All analyses were performed using Statistical Product and Service Solutions (SPSS) 22.0 (IBM Corp, Armonk, NY, USA) software.

## 3. Results

Table 1 presents the descriptive statistics of eHealth literacy, beliefs about medicines and medication adherence. The mean score indicates that the teachers basically had adequate levels of eHealth literacy (all means exceed 2.5) and a low perceived necessity of taking medicines (all means are lower than 2.5). Additionally, they had moderate concerns about taking medicines and sometimes took their medications as prescribed (all means exceed 2.5).

In terms of gender, the independent t test reveals significant differences in specific concern beliefs, and the mean scores reveal that males had stronger concerns about medication than females (see Table 1). Therefore, Hypothesis 1 is partially supported. Regarding age, one-way ANOVA revealed significant differences in functional eHealth literacy, specific necessity beliefs, and medication adherence. The findings of the post hoc comparison reveal that the G1 participants had higher functional eHealth literacy than the G3 participants, that the G3 participants had stronger necessity beliefs about medication than the G1 and G2 participants, and that the G3 participants had better medication adherence than the G2 participants (see Table 2). Hypothesis 2 is also partially supported.

Table 3 indicates that gender, age, eHealth literacy and beliefs about medicines jointly predicted medication adherence, with an explanatory power of 8%, and this result supports Hypothesis 3. Age (*β* = 0.09), functional eHealth literacy (*β* = 0.12) and specific necessity beliefs (*β* = 0.11) positively predicted medication adherence. In addition, specific concern beliefs (*β* = −0.23) negatively predicted medication adherence. These findings indicate that teachers who are older, have higher functional eHealth literacy, and have stronger necessity beliefs about medication tend to take their medications as prescribed. In contrast, those who have stronger concerns about medication are less likely to take their medications as prescribed.

## 4. Discussion

Previous studies have revealed that age plays an important role in eHealth literacy [36,37,38]. The study by Shiferaw et al. [39] observed that health professionals aged 39 or older are more likely to have lower eHealth literacy than health professionals aged 20–29 years. It seems that advanced age is associated with lower levels of eHealth literacy. In this study, we also found this tendency, and our teacher participants aged 42–51 years had lower functional eHealth literacy than teachers aged 22–31 years. This finding implies that despite similar occupations and higher educational levels, age may still be a determinant of an individual’s eHealth literacy. Thus, future studies could design age-specific programs to improve teachers’ eHealth literacy and, in particular, help teachers aged 42–51 years improve their functional eHealth literacy.

This study revealed that male teachers had stronger concerns about medication than female teachers. Teachers aged 42–51 years had stronger necessity beliefs about medication than teachers aged 22–31 years and teachers aged 32–41 years. The findings were consistent with the study of Lu and Zhang [17]. It is inferred that older people are somewhat aware of aging and may feel the need for medication. The reviews by Horne et al. and Shahin et al. [5,40] also showed that adherence is influenced by personal, social, cultural, economic and healthcare system contexts, and that necessity beliefs and concerns are associated with adherence/nonadherence to medicines across a wide range of conditions, medications, and study locations. In general, both the literature and our findings revealed that the gender and age differences in teachers’ beliefs about medicines deserve our attention. It seems that helping male teachers and teachers aged 42–51 years build positive beliefs about their medicines is necessary.

The one-way ANOVA and multiple regression findings showed that young teachers were less likely to take their medications as prescribed than elderly teachers, which is similar to previous studies [19,20]. Recently, scholars and researchers have paid attention to promoting medication adherence in young adults with cancer [41] and sickle cell disease [42] via smartphone applications. Therefore, smartphone applications are likely to improve young teachers’ medication adherence.

Consistent with the studies by Lin et al. [43] and Zisopoulou et al. [44], this study revealed that teachers with higher functional eHealth literacy tended to take their medications as prescribed. Functional eHealth literacy allows individuals to effectively read and understand health information via the internet [31,45]. In Taiwan, the Ministry of Health and Welfare strives to convey correct medical knowledge through health education leaflets, videos, news and the internet [46]. With the ability to search for online information, individuals can use health information to tackle problems and improve their medication adherence [44]. Therefore, teachers with higher functional eHealth literacy tended to have a better understanding of medicine information and, as a result, take their medications as prescribed.

Finally, this study revealed that teachers who had stronger necessity beliefs about medication and who had fewer concerns about medication tended to have greater medication adherence, supporting previous evidence [5,6,7,8,9]. Specific necessity beliefs refer to personal beliefs about the necessity and beneficial effects of taking medicine [4,6]. When individuals believe that medication is necessary, they have better adherence behaviors [9,47], and if they do not, they are likely to have a higher frequency of nonadherence behaviors [18]. Specific concern beliefs refer to personal concerns about the negative effects of one’s medication [4,6]. When individuals are convinced that medications have potential negative effects, they have poor adherence [9,20,47]. Previous studies have shown that when patients’ necessity beliefs about their medications were stronger than their concerns about taking medications, they were more motivated to take medications as prescribed [48,49]. In contrast, when patients’ concerns about their medications outweighed their need for medications, they were more likely to be non-adherers [50].

All of our study participants were teachers who were well educated, and the results of the descriptive statistical analysis show that the teachers had a low perceived necessity of but moderate concerns about taking medication. The results suggest that extra attention should be given and help them voice their concerns and where possible resolve concerns, and allay concerns to shift from emphasis on concerns to necessity. It is suggested to further apply the NCF and Health Belief Model’s perceived barriers and perceived benefits to qualitatively examine those teachers with a low perceived necessity but moderate concerns about taking medication, and to further elicit and address their key beliefs underpinning attitudes and decisions about treatment; for example, how they evaluated treatment necessity and concerns, and negotiated potential side effects, and whether and how factors such as health belief, health condition, peer-to-peer communication is moderated or mediated in regards to medication adherence.

Although this study makes many contributions to the field, it still has several limitations. First, this study was conducted in the form of a self-report survey to investigate the phenomenon that the teachers took medication prescribed by their health care providers (e.g., otorhinolaryngologist, dermatologist, gastrologist, etc.) in the past year. Since the time interval was relatively long, recall bias may exist. Second, we did not collect data on the disease type (e.g., chronic diseases or infectious diseases), symptoms (e.g., diarrhea, common cold, and eye diseases) and other information (e.g., actual medication record), which made this study unable to compare the present instrument with other objective measurements of medication adherence. Moreover, other potential limitations, such as social desirability [51] and unintentional nonadherence bias [52], may also exist. Therefore, caution should be taken when interpreting the results.

In the future, researchers may follow the suggestion of Ryan et al. [53] to collect real-world exposure information, obtain more medications relative to outcomes, and build empirical measurement data. For instance, researchers could ask participants to record the number or frequency of medications taken in compliance with their physician’s instructions and examine the links among gender, age, eHealth literacy, beliefs about medicines and medication adherence for different disease types

Third, caution is needed when generalizing these findings since we used the convenience sampling technique to collect data, and sampling error may occur. For example, we have the risk of recruiting subjects with similar traits and experience. In addition, participants may be more/less involved in the survey based on peer relationships because the questionnaires are distributed through acquaintances, and those who are currently unable to work would be excluded. All of this reduces the generalizability of the study, and we suggest that, if possible, future studies may use random samples to address this issue.

Finally, this cross-sectional study did not provide reliable evidence to support a causal relationship between eHealth literacy and medication adherence. As medication adherence is multifaceted and intertwined with many individual, social and cultural factors, other factors that might influence beliefs about medication are cognitive knowledge and perception about illness and emotional responses to the illness. These cognitive knowledge and illness perceptions might activate their behavioral actions of medication adherence. Further research is required to fully identify the associations between religious beliefs, control beliefs and illness knowledge and medication adherence.

## 5. Conclusions

The present study is the first to identify the relationships among gender, age, eHealth literacy (including the functional and proactive roles of interactive and critical eHealth literacy), beliefs about medicines, and medication adherence among elementary and secondary school teachers. Although the findings differed from our expectations, we did not find an association of interactive and critical eHealth literacy with medication. This study still provides a contribution to the academic evidence and forms a basis for future study.

This study found that male teachers had stronger concerns about medication than female teachers. Teachers aged 42–51 years had lower functional eHealth literacy and stronger necessity beliefs about medication than teachers aged 22–31 years. Finally, teachers who were older, had higher functional eHealth literacy, had stronger necessity beliefs about medication, and had fewer concerns about medication tended to have greater medication adherence.

Previous studies have demonstrated that medication adherence is influenced by a number of issues, including side effects, cost of the medication, dosing frequency, routes of administration, patient beliefs, demographics and comorbidities [21,54]. Future studies may consider realizing the chronic medical conditions of the study population, which we did not dig into. Moreover, our results suggest that helping teachers develop high eHealth literacy and positive beliefs about medicines is an effective strategy for improving medication adherence. More teachers with proper health literacy could influence more students to have proper health literacy and medication adherence concepts. Therefore, refining the design of the program for teachers to have better health-promoting behavior relies on the awareness of the differences. Health programs for teachers should focus on helping them have a better understanding of the potential negative effects of medication, reduce concerns about medication, and further improve their medication adherence.

## Figures and Tables

**Table 1 ijerph-19-06926-t001:** Overview of the mean (and standard deviation in parentheses) and max (and min in parentheses) values and independent t test results of the participant characteristics, with stratification by gender.

	Gender	Total (*n* = 485)	Male (*n* = 137)		Female (*n* = 348)		*t* Value	*p* Value
Variables		*M* (*SD*)	*M* (*SD*)	*Max* (*Min*)	*M* (*SD*)	*Max* (*Min*)		
Functional eHealth literacy		3.90 (0.74)	3.95 (0.78)	5.00 (1.33)	3.87 (0.72)	5.00 (1.00)	1.03	0.303
Interactive eHealth literacy		3.93 (0.64)	3.90 (0.68)	5.00 (1.00)	3.94 (0.62)	5.00 (1.00)	−0.56	0.578
Critical eHealth literacy		3.86 (0.67)	3.87 (0.74)	5.00 (1.20)	3.85 (0.64)	5.00 (1.00)	0.20	0.840
Specific necessity beliefs		1.95 (0.76)	2.05 (0.81)	4.75 (1.00)	1.91 (0.74)	4.75 (1.00)	1.72	0.086
Specific concerns beliefs		3.00 (0.89)	3.13 (0.88)	5.00 (1.00)	2.95 (0.89)	5.00 (1.00)	2.00	0.046
Medication adherence		3.36 (0.63)	3.34 (0.67)	5.00 (1.00)	3.37 (0.62)	4.88 (1.00)	−0.58	0.564

**Table 2 ijerph-19-06926-t002:** Overview of the mean (and standard deviation in parentheses) and max (and min in parentheses) values and one-way ANOVA results of the participant characteristics, with stratification by age.

	Age Groups	G1 (*n* = 70) 22–31 yrs		G2 (*n* = 211) 32–41 yrs		G3 (*n* = 204) 42–51 yrs		*F* *-*	*p* *-*	Post
Variables		*M* (*SD*)	*Max* (*Min*)	*M* (*SD)*	*Max* *(Min*)	*M* (*SD*)	*Max* (*Min*)	Value	Value	Hoc
Functional eHealth literacy		4.07 (0.67)	5.00 (2.00)	3.92 (0.68)	5.00 (2.00)	3.82 (0.81)	5.00 (1.00)	3.18	0.042	G1 > G3
Interactive eHealth literacy		4.02 (0.67)	5.00 (1.00)	3.95 (0.57)	5.00 (2.00)	3.88 (0.69)	5.00 (1.00)	1.50	0.225	
Critical eHealth literacy		3.96 (0.82)	5.00 (1.00)	3.84 (0.61)	5.00 (1.40)	3.84 (0.68)	5.00 (1.20)	0.92	0.398	
Specific necessity beliefs		1.84 (0.67)	3.50 (1.00)	1.83 (0.66)	4.50 (1.00)	2.11 (0.85)	4.75 (1.00)	Welch 7.63	0.001	G3 > G1, G2
Specific concerns beliefs		2.97 (0.86)	4.50 (1.00)	3.00 (0.90)	5.00 (1.00)	3.01 (0.89)	5.00 (1.00)	0.040	0.960	
Medication adherence		3.32 (0.68)	4.38 (1.00)	3.28 (0.60)	4.88 (1.00)	3.46 (0.64)	5.00 (1.38)	4.35	0.013	G3 > G2

Note: Since specific necessity beliefs did not meet the homogeneity of variance assumption, we used the Welch test and the Games–Howell method to perform the analysis.

**Table 3 ijerph-19-06926-t003:** Multiple regression model (enter method) using gender, age, eHealth literacy and beliefs about medicine as independent variables.

Variables	Medication Adherence (Dependent Variable)					
	B	SE	*β*	*p*-Value	Tolerance	VIF
Gender (Male = 0, Female = 1)	0.04	0.06	0.03	0.557	0.98	1.02
Age	0.01	0.00	0.09	0.039	0.96	1.05
Functional eHealth literacy	0.10	0.04	0.12	0.013	0.85	1.18
Interactive eHealth literacy	−0.11	0.06	−0.11	0.081	0.52	1.94
Critical eHealth literacy	0.09	0.06	0.09	0.121	0.55	1.81
Specific necessity beliefs	0.09	0.04	0.11	0.013	0.94	1.06
Specific concern beliefs	−0.16	0.03	−0.23	<0.001	0.97	1.04
*F* = 6.67, *p* < 0.001, *R*^2^ = 0.09, adjusted *R*^2^ = 0.08						

VIF: Variance inflation factor.

## Data Availability

The dataset cannot be reused or provided since our participants agreed to their data being used only for this study.

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
