# Peer review of "The Associations among Gender, Age, eHealth Literacy, Beliefs about Medicines and Medication Adherence among Elementary and Secondary School Teachers"

_ijerph, 2022, doi:10.3390/ijerph19116926_

Round 1

Reviewer 1 Report

In this manuscript, the authors investigate relationships between health literacy, beliefs about medicines and medication adherence. Although I think this manuscript has merit, I would like the authors to address the following comments:

My main comment relates to the sample that is chosen for this study. In the abstract, the authors state that most research on medication adherence has been carried out in "chronic disease patient samples rather than normal samples". In the introduction, (at the bottom of page 2), they further explain their choice for a teacher sample, but I feel their is still information mission that would justify their choice of research sample/population:
    - It is unclear to me if their choice is related to the advantages of conducting a study on treatment adherence in a general population sample (therefore removing potential biases related to individuals consulting in (specialty) clinics, or whether the behavior of teachers would merit specific research (i.e. Do the authors expect teachers behave differently from the general population or are there different consequences of their behaviors.
    - Related: there is no discussion in the manuscript of potential sampling biases related to the choice of research sample: i.e. Teachers may be more/less compelled to participate compared to samples from other populations, the sample would exclude teachers who are currently unable to work, …
    - There is very limited information in the paper on the types of (chronic) medical conditions the teachers in the sample are having, or what type of medication they are taking. Were participants excluded if they did not take any chronic medication? As treatment adherence and medication beliefs can greatly differ based on conditions and specific medications, there would be a need to expand the manuscript with this information (see also comment on the BMS below)

The abstract only mentions health literacy in passing, whereas the introduction is starting with a lot of information on health literacy. This feels as a mismatch.

When discussing studies on adherence and medication beliefs (p2, second paragraph), I feel it would make sense to integrate findings on a systematic review on this topic: https://journals.plos.org/plosone/article?id=10.1371/journal.pone.0080633

Methods:
2.2.2. in questionnaires like the BMQ, specific medication beliefs are investigated in reference to a specific medication. How does this work in the BMS questionnaire? How did the authors deal with participants taking more that one medication?

Similarly, for the medication adherence scale, it would be interesting to know what this instrument looked like, and how the authors dealt with participants taking no medications or more than one type of medication.

In general, there is very little information on the sample (both in the methods section and results section) that would make it possible to compare this sample to other samples on which information related to treatment adherence has been conducted.

Data collection: it is not clear what the 80.8 % of valid data is referring to: is this response rate? Missing data? Please expand on this.

In the discussion, the Marshall et al. systematic review is mentioned, but is is misrepresented in that this review deals specifically with cancer patients. At this point, adding and integrating information from a more general systematic review (i.e. The Horne et al. review mentioned above) would be beneficial.

Ethics: the ethics statement indicates that "Ethical approval for this study was waived by Taiwan Centers for Disease Control Policy # 1010265075 because this study was conducted in a general teaching environment for educational purposes and all subjects were voluntary participants."
I am not sure if the study described fits this purpose (educational purposes). Could the authors expand on their thought process related to this (or show similar decisions by ethical committees showing that they are exempt from ethical review? 

Author Response

Comment 1

My main comment relates to the sample that is chosen for this study. In the abstract, the authors state that most research on medication adherence has been carried out in "chronic disease patient samples rather than normal samples". In the introduction, (at the bottom of page 2), they further explain their choice for a teacher sample, but I feel there is still information mission that would justify their choice of research sample/population:

It is unclear to me if their choice is related to the advantages of conducting a study on treatment adherence in a general population sample (therefore removing potential biases related to individuals consulting in (specialty) clinics, or whether the behavior of teachers would merit specific research (i.e. Do the authors expect teachers behave differently from the general population or are there different consequences of their behaviors.

Reply 1:

We selected teachers as a target group not only for academic reasons (bridging the gap of literature scarcity) but also for educational practice. Taiwanese school-aged children spent 8-9 hours a day in schools, and teachers became critical life instructors and mentors during that time. In terms of teaching health education, teachers are not only the main body of school health efforts but also role models for students' healthy behavior. Thus, we suggest that improving teachers' health-promoting behaviors will induce students' healthy behaviors and produce positive health outcomes. As prior studies have supported, educator health literacy and the ability to support student health are associated with school teacher-level characteristics (Mansfield et al., 2021; Peterson et al., 2001); therefore, to provide effective assistance, it is important to understand differences in e-health literacy beliefs among primary and secondary school teachers. In addition, E-health literacy is defined as the ability to obtain needed and correct health information through electronic resources and further use this knowledge to make health decisions or resolve health-related problems (Norman & Skinner, 2006). Considering that teachers are well educated, we expect that they may have better health literacy than the public and thus may have dissimilar attitudes/behaviors toward medication. Therefore, it is interesting to take teachers as a target sample to explore the relationship among eHealth literacy, beliefs about medicines and medication adherence.

References mentioned in the reply

Mansfield R, Humphrey N, Patalay P. Educators' perceived mental health literacy and capacity to support students' mental health: associations with school-level characteristics and provision in England. Health Promot Int. 2021 Dec 23;36(6):1621-1632. doi: 10.1093/heapro/daab010. PMID: 33667299; PMCID: PMC8699399.

Peterson FL, Cooper RJ, Laird JM. Enhancing teacher health literacy in school health promotion: a vision for the new millennium. J Sch Health. 2001 Apr;71(4):138-44. doi: 10.1111/j.1746-1561.2001.tb01311.x. PMID: 11354982.

Norman CD, Skinner HA. eHealth literacy: essential skills for consumer health in a networked world. J Med Internet Res 2006 Jun 16;8(2):e9. doi: 10.2196/jmir.8.2.e9

Comment 2

Related: there is no discussion in the manuscript of potential sampling biases related to the choice of research sample: i.e. Teachers may be more/less compelled to participate compared to samples from other populations, the sample would exclude teachers who are currently unable to work, …

There is very limited information in the paper on the types of (chronic) medical conditions the teachers in the sample are having, or what type of medication they are taking. Were participants excluded if they did not take any chronic medication? As treatment adherence and medication beliefs can greatly differ based on conditions and specific medications, there would be a need to expand the manuscript with this information (see also comment on the BMS below)

Reply 2:

We appreciate the reviewer’s comment.

        Regarding the sampling bias, to make readers more cautious about the results of this study, we discuss the possible limitations of the convenience sampling method used in this study; please see the discussion section. In addition, all possible participants were included in this investigation, although some did not take any chronic medication and we added this information in this revision. The reason is that this is the first study to examine the teacher population, as a pilot study, to form a basis for future study and give possible suggestions to enhance teachers’ health-promoting behaviors. Therefore, we sought to obtain information in a relatively comprehensive manner, rather than limiting it to those who have specific or chronic diseases, to better understand the current state of teachers’ adherence, medication beliefs, and eHealth literacy. However, we understand that studies have demonstrated that medication adherence is influenced by a number of issues, including side effects, cost of the medication, dosing frequency, routes of administration, patient beliefs, demographics and comorbidities (e.g., Chan, Cooper, Lycett & Horne, 2020; Gast & Mathes, 2019), and agree it is an important issue to realize the chronic medical conditions of the study population. A relevant study will be conducted in the future.

References mentioned in the reply

Gast, A., Mathes, T. Medication adherence influencing factors—an (updated) overview of systematic reviews. Syst Rev 8, 112 (2019). https://doi.org/10.1186/s13643-019-1014-8

Chan AHY, Cooper V, Lycett H and Horne R (2020) Practical Barriers to Medication Adherence: What Do Current Self- or Observer-Reported Instruments Assess? Front. Pharmacol. 11:572. doi: 10.3389/fphar.2020.00572

Comment 3

The abstract only mentions health literacy in passing, whereas the introduction is starting with a lot of information on health literacy. This feels as a mismatch.

Reply 3:

We appreciate the reviewer’s comment. We have added a description of health literacy to make it more appropriate. Regarding the modification, please refer to the description in blue as follows:

Abstract: Background: A lack of health literacy may negatively impact patient adherence behavior in health care delivery, leading to a major threat to individual health and well-being and an increasing financial burden on national healthcare systems. Therefore, how to cultivate citizens' health literacy, especially e-health literacy that is closely related to the Internet, may be seen as a way to reduce the financial burden of the national healthcare systems, which is the responsibility of every citizen. However, previous studies on medication adherence have mostly been conducted with chronic disease patient samples rather than normal samples. Teachers are not only the main body of school health efforts but also role models for students' healthy behavior. Therefore, understanding differences in e-health literacy beliefs among schoolteachers would be helpful for improving the existing health promoting programs and merit specific research. Aims: The present study identified the relationships among gender, age, electronic health (eHealth) literacy, beliefs about medicines, and medication adherence among elementary and secondary school teachers. Methods: A total of 485 teachers aged 22-51 years completed a pen-and-paper questionnaire. The instrument included an eHealth literacy scale, a belief about medicines scale and a medication adherence scale. Results: The results showed a significant difference between genders in necessity beliefs about medication (t = 2.00, p < .05) and a significant difference between ages in functional eHealth literacy (F = 3.18, p < .05) and in necessity beliefs about medication (Welch = 7.63, p < .01). Moreover, age (β = .09), functional eHealth literacy (β = .12), and necessity beliefs about medication (β = .11) positively predicted medication adherence, while concerns about medication (β = -.23) negatively predicted medication adherence. Conclusions: The results showed that male teachers had stronger concerns about medication than female teachers. Teachers aged 42-51 years had lower functional eHealth literacy and stronger necessity beliefs about medication than teachers aged 22-31 years. In addition, teachers who were older, had higher functional eHealth literacy, had stronger necessity beliefs about medication, and had fewer concerns about medication tended to take their medications as prescribed. These findings revealed that helping teachers develop high eHealth literacy and positive beliefs about medicines is an effective strategy for improving medication adherence.

Comment 4

When discussing studies on adherence and medication beliefs (p2, second paragraph), I feel it would make sense to integrate findings on a systematic review on this topic: https://journals.plos.org/plosone/article?id=10.1371/journal.pone.0080633

Reply 4:

Thank you for the suggestions. We have integrated Horne et al.’s meta-analytic review of the necessity-concerns framework for understanding patients’ adherence-related beliefs about medicines prescribed for long-term conditions and related references. Regarding the revision, please see the following content:

Medication adherence is usually defined as whether patients follow the prescription to take medicine, accept therapy, and continuously take a prescribed medication until they are cured or stop when the condition is sufficiently improved (1). Patient adherence in healthcare services affects personal health and well-being, as well as the financial affordability of the national healthcare system (2). Patient adherence can be seen as a way to reduce the financial burden on the national health care system and help the country's medical economy. However, patients’ medication adherence is intertwined with many factors, including their medication beliefs, which are defined as an individual’s beliefs about the necessity of and problems with medication (3, 4).

Horne et al.’s meta-analytic review of the necessity-concerns framework for understanding patients’ adherence-related beliefs about medicines prescribed for long-term conditions pointed out that the ‘Necessity-Concerns Framework (NCF)’ potentially offers a convenient model for understanding patients’ perspectives on prescribed medicines, including perceptions of personal need for treatment (Necessity beliefs) and Concerns about a range of potential adverse consequences (5). Horne et al. (2013) mentioned that nonadherence is often a hidden problem that patients may not be willing to express doubts or concerns about prescribed medication and to report nonadherence sometimes because they fear that this will be perceived by the prescriber as a lack of faith in them (5). Prior studies have shown that greater adherence is linked to fewer concerns about medication and stronger necessity beliefs about medication (6-9). Thus, patients with positive perspectives on medicines are more likely than those with negative perspectives to take their medications as prescribed.

References mentioned in the reply

  1. Ho, P. M.; Bryson, C. L.; Rumsfeld, J. S. Medication adherence: its importance in cardiovascular outcomes. Circulation 2009, 119(23), 3028-3035.
  2. Wu, D.; Lowry, P. B.; Zhang, D.; Parks, R. F. Patients’ compliance behavior in a personalized mobile patient education system (PMPES) setting: Rational, social, or personal choices? J. Med. Inform. 2021, 145, 104295.
  3. Horne, R.; Weinman, J.; Hankins, M. The beliefs about medicines questionnaire: the development and evaluation of a new method for assessing the cognitive representation of medication. Health 1999, 14(1), 1-24.
  4. Foot, H.; La Caze, A.; Gujral, G.; Cottrell, N. The necessity–concerns framework predicts adherence to medication in multiple illness conditions: A meta-analysis. Educ. Couns. 2016, 99(5), 706-717.
  5. Horne, R.; Chapman, S. C.; Parham, R.; Freemantle, N.; Forbes, A.; Cooper, V. Understanding patients’ adherence-related beliefs about medicines prescribed for long-term conditions: a meta-analytic review of the Necessity-Concerns Framework. PloS one 2013, 8(12), e80633.
  6. Kalichman, S.; Kalichman, M. O.; Cherry, C. Medication beliefs and structural barriers to treatment adherence among people living with HIV infection. Health 2016, 31(4), 383-395.
  7. Kalichman, S. C.; Eaton, L.; Kalichman, M. O.; Cherry, C. Medication beliefs mediate the association between medical mistrust and antiretroviral adherence among African Americans living with HIV/AIDS. HealthPsychol. 2017, 22(3), 269-279.
  8. Horne, R.; Albert, A.; Boone, C. Relationship between beliefs about medicines, adherence to treatment, and disease activity in patients with rheumatoid arthritis under subcutaneous anti-TNFα therapy. Patient Prefer. 2018, 12, 1099.
  9. Cea-Calvo, L.; Marín-Jiménez, I.; de Toro, J.; Fuster-RuizdeApodaca, M. J.; Fernández, G.; Sánchez-Vega, N.; Orozco-Beltrán, D. Association between non-adherence behaviors, patients’ experience with healthcare and beliefs in medications: A survey of patients with different chronic conditions. Med. Res. Opin. 2020, 36(2), 293-300.

Comment 5-1

Methods:
2.2.2. in questionnaires like the BMQ, specific medication beliefs are investigated in reference to a specific medication. How does this work in the BMS questionnaire? How did the authors deal with participants taking more than one medication? Similarly, for the medication adherence scale, it would be interesting to know what this instrument looked like, and how the authors dealt with participants taking no medications or more than one type of medication.

Reply 5-1

Thank you for the comments. Yes, specific medication belief refers to medicine prescribed for a specific illness in the BMQ. However, as Horne et al. (1999) reported, the BMQ was intended to assess commonly held beliefs about medicines. In addition, they found that the use of six disease samples obtained about the same factor structure, which represented the core themes underpinning common representations of specific and general medication. This means that, regardless of the disease type, specific beliefs can be distinguished from general beliefs. Since we focus on the specific beliefs in the general population, we followed the structure of specific beliefs in the BMQ to design the BMS. The items of the BMS and MAS are also provided in the appendix in this revision for your reference. In our study, we focus on the beliefs about the medicine prescribed by the doctor and the medication adherence for these kinds of medicines and did not address the type, amount and the indications-and-usage of the medicine that participants took at the same time (both in the BMS and MAS). Nevertheless, we do value your suggestion and will incorporate it into our follow-up study.

References mentioned in the reply

Horne, R.; Weinman, J.; Hankins, M. The beliefs about medicines questionnaire: the development and evaluation of a new method for assessing the cognitive representation of medication. Psychol. Health 1999, 14(1), 1-24.

Comment 5-2

--In general, there is very little information on the sample (both in the methods section and results section) that would make it possible to compare this sample to other samples on which information related to treatment adherence has been conducted.

--Data collection: it is not clear what the 80.8 % of valid data is referring to: is this response rate? Missing data? Please expand on this.

--In the discussion, the Marshall et al. systematic review is mentioned, but is is misrepresented in that this review deals specifically with cancer patients. At this point, adding and integrating information from a more general systematic review (i.e. The Horne et al. review mentioned above) would be beneficial.

Reply 5-2

Thank you for the comments. We apologize for the unclear description and information in the manuscript. However, for your concern, please allow us to further explain. The aim of this study is to provide a more complete understanding of the teacher population and therefore to provide appropriate suggestions for health promotion programs. Thus, the extant information about our sample should be sufficient because making comparisons between different populations is not the goal. To make this information clearer to the reader, we added the relevant information in this revision. In addition, the main concern of this study was participants' beliefs/adherence about the medicine prescribed by the doctor rather than over the-counter medicine. Meanwhile, considering that this is a pilot study, we believe that examining medication attitudes (beliefs and adherence) in a more general way would be more appropriate, so we did not limit the investigation to one disease/treatment. We appreciate your comments and hope that the clarifications and revisions have improved the quality of the current manuscript.

For the unclear description of “80.8% of valid data”, we expanded it as follows: Finally, a convenience sample of six hundred questionnaires was distributed, and 485 questionnaires were returned, resulting in a response rate of 80.8%.

Regarding the misrepresentations of Marshall et al.’s systematic review in the discussion section, we have modified and added a more general systematic review of Horne et al. and Shahin et al. to make the arguments more solid and logical. For example, we have made the following changes:

This study revealed that male teachers had stronger concerns about medication than female teachers. Teachers aged 42-51 years had stronger necessity beliefs about medication than teachers aged 22-31 years and teachers aged 32-41 years. The findings were consistent with the study of Lu and Zhang (17), who found that older patients and females were more willing to take medications. It is inferred that older people are somewhat aware of aging and may feel the need for medication. Horne et al.’s and Shahin et al.’s review (5, 40) also showed that adherence is influenced by personal, social, cultural, economic and healthcare system contexts and that necessity beliefs and concerns are associated with adherence/nonadherence to medicines across a wide range of conditions, medications, and study locations. In general, both the literature and our findings revealed that the gender and age differences in teachers’ beliefs about medicines deserve our attention. It seems that helping male teachers and teachers aged 42-51 years build positive beliefs about their medicines is necessary.

Shahin, W.; Kennedy, G. A.; Stupans, I. The impact of personal and cultural beliefs on medication adherence of patients with chronic illnesses: A systematic review. Patient Prefer. Adher.2019, 13, 1019.

Comment 5-3
Ethics: the ethics statement indicates that "Ethical approval for this study was waived by Taiwan Centers for Disease Control Policy # 1010265075 because this study was conducted in a general teaching environment for educational purposes and all subjects were voluntary participants."
I am not sure if the study described fits this purpose (educational purposes). Could the authors expand on their thought process related to this (or show similar decisions by ethical committees showing that they are exempt from ethical review? 

Reply 5-3

Thank you for the comments. In view of personal privacy and anonymous surveys, the first page of our questionnaire states that filling in the questionnaire and submitting it is deemed consent. In addition, the first page of the questionnaire contained the contact information of the research team, and the participants could contact us if they had questions or needed help in case of confusion when completing the questionnaire. The survey instructions also clearly informed the teacher-participants of the purposes of the research and of their rights regarding joining or dropping out of this study. Teachers who are interested in this study have plenty of time to think and decide whether to participate in this investigation after reading the research purposes and related information on the front page of the survey. Those who would like to engage in this study can return the questionnaire on the agreed time. At the same time, two checkpoints were provided in the research design so that the respondents could stop at any time: (1) As they were completing the survey, the respondents could quit at any time. (2) Upon completing the entire survey, the respondents determined whether to hand it in. The respondents were assured that their participation was voluntary, anonymous, and strictly confidential and that they had the right to refuse to participate in the study at any time without any penalty. In addition, the teachers who participated in the study received a gift, regardless of whether they completed the questionnaire.

Reviewer 2 Report

The article is interesting, and the researched problem has scientific potential. However, some problems need to be solved:

  1. The abstract must be restructured in a synthetic manner.

  1. Literature review must be extended. Literature review needs to study more resource on particularly on eHealth literacy.

  1. The authors must provide (eventually in the appendix) the structure of the questionnaire used.

  1. Data processing is performed using descriptive statistics. The article would gain value if complex statistical methods were used to establish the relationships between variables (SEM, MANOVA etc.)

  1. The discussion or conclusions section must include theoretical and managerial implications, research limitations, and future research directions would be helpful.

The article presents some scientific value and can be published after a careful review of the reported issues.

Author Response

Reply

We appreciate the reviewer’s constructive comments.

        The abstract was reorganized in a more synthetic manner. For the literature, we have elaborated eHealth literacy and integrated Horne et al.’s meta-analytic review of the necessity-concerns framework for understanding patients’ adherence-related beliefs about medicines prescribed for long-term conditions and related references. The questionnaire used in this study was also provided in this revision; please refer to the appendix.

        Regarding the use of advanced statistical analysis methods, considering that independent t tests, ANOVA and regressions have been able to adequately test research hypotheses in the present study and many scholars (e.g., Lyu, 2020) suggested when choosing a data analysis method, a quantitative technique that is relatively simple and easy to implement should be selected under the condition that the research objective can be effectively accomplished. Therefore, we selected these three statistical methods to address our data. Based on your suggestion, we will launch another paper by performing a complex statistical method to address the deep relationships among variables. Finally, we include theoretical and managerial implications, research limitations, and future research directions in this revision.

References mentioned in the reply

Lyu J. The quantitative methods in education empirical research in China: A review on five years’ application. Journal of East China Normal University (Educational Sciences). (2020) 9:36-55. doi: 10.16382/j.cnki.1000-5560.2020.09.003 (in Chinese)

Round 2

Reviewer 2 Report

The paper can be published in current form.